# DOT: Gene-set analysis by combining decorrelated association statistics

Olga A. Vsevolozhskaya[1], Min Shi[2], Fengjiao Hu[2], Dmitri V. Zaykin[2]*

**1** Department of Biostatistics, College of Public Health, University of Kentucky, Lexington, Kentucky, United States of America, **2** Biostatistics and Computational Biology, National Institute of Environmental Health Sciences, National Institutes of Health, Research Triangle Park, North Carolina, United States of America

* dmitri.zaykin@nih.gov

**Data Availability Statement:** The URL for software referenced in this article is available at: https://github.com/dmitri-zaykin/Total_Decor.

**Funding:** This research was supported in part by the Intramural Research Program of the National

## Abstract

Historically, the majority of statistical association methods have been designed assuming availability of SNP-level information. However, modern genetic and sequencing data present new challenges to access and sharing of genotype-phenotype datasets, including cost of management, difficulties in consolidation of records across research groups, etc. These issues make methods based on SNP-level summary statistics particularly appealing. The most common form of combining statistics is a sum of SNP-level squared scores, possibly weighted, as in burden tests for rare variants. The overall significance of the resulting statistic is evaluated using its distribution under the null hypothesis. Here, we demonstrate that this basic approach can be substantially improved by decorrelating scores prior to their addition, resulting in remarkable power gains in situations that are most commonly encountered in practice; namely, under heterogeneity of effect sizes and diversity between pairwise LD. In these situations, the power of the traditional test, based on the added squared scores, quickly reaches a ceiling, as the number of variants increases. Thus, the traditional approach does not benefit from information potentially contained in any additional SNPs, while our decorrelation by orthogonal transformation (DOT) method yields steady gain in power. We present theoretical and computational analyses of both approaches, and reveal causes behind sometimes dramatic difference in their respective powers. We showcase DOT by analyzing breast cancer and cleft lip data, in which our method strengthened levels of previously reported associations and implied the possibility of multiple new alleles that jointly confer disease risk.

## Author summary

Joint analysis of association between the outcome and a group of SNPs within a genetic region is increasingly recognized to complement single-SNP analysis and shed light on the underlying molecular mechanisms. However, the correlation among GWAS association results calls for specifically tailored statistical methods. Here we propose DOT (Decorrelation by Orthogonal Transformation) method that can efficiently combine evidence of association over different SNPs and genes within a pathway without access to the

Institutes of Health (NIH), National Institute of Environmental Health Sciences. The funders had no role in study design, data collection and analysis, decision to publish, or preparation of the manuscript.

**Competing interests:** The authors have declared that no competing interests exist.

original genotypic data. DOT is fast, does not rely on a permutation algorithm, and is often dramatically more powerful than other popular methods, such as VEGAS and the recently proposed ACAT. We believe that DOT will become a useful addition to the toolbox of methods based on the summary statistics for the GWAS community.

This is a *PLOS Computational Biology* Methods paper.

## Introduction

During the recent years, genome-wide association studies (GWAS) uncovered a wealth of genetic susceptibility variants. The emergence of new statistical approaches for the analysis of GWAS have largely contributed to that success. The majority of these methods require access to individual-level data, yet methods that require only summary statistics have been developed as well. The rising popularity of summary-based methods for the analysis of genetic associations has been motivated by many factors, among which is convenience and availability of summary statistics and high statistical power that can often match the power of analysis based on individual records [1–3].

   Many types of association tests, including those originally developed for individual-level records, can be presented in terms of added summary statistics. For example, gene set analysis (GSA) tests or burden and overdispersion tests for rare variants [2, 4, 5], can be written as a weighted sum of summary statistics. In GSA applications, methods based on combined summary statistics can be used to efficiently aggregate information across many potentially associated variants within individual genes, as well as over several genes that may represent a common etiological pathway. When within-gene association statistics (or equivalently, P-values) are being combined, linkage disequilibrium (LD) needs to be accounted for, because LD induces correlation among statistics. The correlation among association test statistics for individual SNPs without covariates is the same as the correlation between alleles at the corresponding SNPs, if the genotype-phenotype relationship is linear. This fact allows one to model a set of statistics using a multivariate normal (MVN) distribution with the correlation matrix equal to the matrix of LD correlations. More generally, in the presence of covariates correlated with SNPs, MVN correlations among association statistics will depend not only on LD but also on other covariates in the model [6, 7].

   When SNPs are coded as 0,1,2 values, reflecting the number of copies of the minor allele, the LD matrix of correlations can be obtained from SNP data as the sample correlation matrix. It can also be directly estimated from haplotype frequencies whenever those are available or reported. Specifically, the LD (i.e., the covariance between alleles $i$ and $j$; $D_{ij}$) is defined by the difference between the di-locus haplotype frequency, $P_{ij}$, and the product of the frequencies of two alleles, $D_{ij} = P_{ij} - p_i p_j$. Then, the correlation between a pair of SNPs is defined as $r_{ij} = \frac{D_{ij}}{\sqrt{p_i(1-p_i)p_j(1-p_j)}}$. The di-locus $P_{ij}$ frequency is defined as the sum of frequencies of those haplotypes that carry both of the minor alleles for SNPs $i$ and $j$. Similarly, $p_i$ allele frequency is the sum of haplotype frequencies that carry the minor allele of SNP $i$.

   It is important to distinguish situations, in which the LD matrix is estimated using the same data that was used to compute the association statistics from those, where the estimated LD matrix is obtained based on a suitable population reference panel. The reference panel approach is implemented in popular web-based association analysis platforms, such as "VEGAS" [8] or "Pascal" [9]. Based on a user-provided list of $L$ SNPs, with the corresponding

association P-values, VEGAS queries an online reference panel resource to obtain the matrix of LD correlations. P-values are then transformed to normal scores $P_i \rightarrow Z_i$, $i = 1, \ldots, L$, and vector $\mathbf{Z}$ is assumed to follow zero-mean MVN distribution under the null hypothesis of no association. The individual statistics in VEGAS are then combined as $\mathrm{TQ} = \sum_{i=1}^{L} Z_i^2$, (where TQ stands for "Test by Quadratic form") and the overall SNP-set P-value is derived empirically by simulating a large number ($j = 1, \ldots, B$) of zero-mean MVN vectors, adding their squared values to obtain statistics $\mathrm{TQ}^{(j)}$ and computing the proportion of times when $\mathrm{TQ}^{(j)} > \mathrm{TQ}$. The statistics similar to TQ are ubiquitous and appear in many proposed tests that aggregate association signals within a genetic region.

As exemplified by VEGAS, the distribution of TQ must explicitly incorporate LD. However, an alternative approach that implicitly incorporates LD can be based on first decorrelating the association summary statistics, and then exploiting the resulting independence to evaluate the distribution of the sum of decorrelated statistics, which we call Decorrelation by Orthogonal Transformation (DOT). This general idea is straightforward and have been used in many contexts, including methods that utilize individual records [10]. For instance, Zaykin et al. suggested a variation of this approach for combining P-values (or summary statistics) but have not studied power properties of the method in detail [11].

Here, we propose a new decorrelation-based method for combining single-SNP summary association statistics. We derive theoretical properties of our method and explore asymptotic power of both DOT and TQ type of statistics. To the best of our knowledge, we are the first ones to derive the asymptotic distributions of DOT and TQ under the alternative hypothesis. Our results show that decorrelation can provide surprisingly large power boost in biologically realistic scenarios. However, high statistical power is not the only advantage of the proposed framework. Once statistics are decorrelated, one can tap into a wealth of powerful methods developed for combining independent statistics. These methods, among others, include approaches that emphasize the strongest signals by combining the top-ranked results [11–16].

Our theoretical analyses also reveal an unexpected result, showing that in many practical settings tests based on the statistic TQ do not gain power with the increase in $L$ (assuming the same pattern of effect sizes for different values of $L$), while the proposed method steadily gains power under the same conditions. Specifically, the proposed decorrelation method gains power when the effect sizes and/or pairwise LD values become increasingly more heterogeneous. The reasons behind the respective behaviors of tests based on TQ and DOT are explored here theoretically and confirmed via simulations. We further derive power approximations that are useful for understanding power properties of the studied methods.

To showcase our method, we evaluate associations between breast cancer susceptibility and SNPs in estrogen receptor alpha (*ESR1*), fibroblast growth factor receptor 2 (*FGFR2*), RAD51 homolog B (*RAD51B*), and TOX high mobility group box family member 3 (*TOX3*) genes, without access to raw genotype data. We first test for a joint association between SNPs in those four genes and breast cancer risk by decorrelating summary statistics based on the overall LD gene structure. We then describe how to follow up on the joint association results and identify one or more SNPs that drive joint association with disease risk. To further validate the utility of DOT, we also applied it to summary statistics of a recent GWAS of cleft lip with and without cleft palate. Both of our real data analyses confirmed previous associations and revealed new associations, suggesting new potential breast cancer and cleft lip SNP markers.

## Results

As an introductory example of power analysis, we considered two simulated SNPs and a linear regression model $Y = \boldsymbol{\beta} \mathbf{X} + \epsilon$, where $X$ has a bivariate normal distribution, $\boldsymbol{\beta} = \{0.3, 0\}$, and $\epsilon$

has a Laplace distribution with unit variance. Thus, in this model $Y$ does not have a normal distribution, however we expect that the theoretical powers for TQ and DOT tests, as derived in "Materials and Methods" section, will match the empirical power. We assumed sample size of 500. In the first simulation experiment with 10,000 simulated regressions, we assumed the bivariate correlation $R = 0.99$. Although two $\beta$ coefficients are distinct, the mean values of association statistics induced by this model are similar to each other and they both are approximately equal to 0.29. These values can be obtained via Eq 2. Our noncentrality analysis in that section suggests that similarity of the mean values may lead to power advantage of the test TQ. The respective powers of the two tests were 0.87 and 0.80, empirically, and 0.86 and 0.80 by the theoretical calculation. In the second simulation experiment, we lowered $R$ to 0.5. This caused the mean values to become distinct (0.29 and 0.14) and this difference of the two means caused the order of power to change, in agreement with our theoretical analysis. Powers now became 0.72 and 0.80, for TQ and DOT, respectively. In this case, empirical and theoretical powers matched to two digits. There is still difference in power at $R = 0.2$ (0.75 vs. 0.80), but of course, in the case $R = 0$, the two methods are identical. The power of DOT here is constant, and this reflects a special case, when only a single SNP has a non-zero effect size and, in addition, all correlations between SNPs are the same. We provide R software script which can not only reproduce these results, but is also capable of power analysis with larger correlation matrices, i.e., cases with multiple SNPs. Correlation matrices are generated as symmetric matrices of random numbers and then converted to positive definite ones using the package "Matrix" [17]. Using this script, we evaluated the type-I error of both methods, assuming $\alpha$-level 0.05, 10 SNPs, and $\boldsymbol{\beta} = 0$. We found the type-I error to be close to the nominal level, using 100,000 simulations (0.04815 for DOT and 0.05002 for Tq). We note that the calculations are very fast and that the 100,000 simulation runs were completed in less than ten minutes on a typical laptop.

Further, we conducted a different set of extensive simulation experiments to study statistical power of the proposed method based on the decorrelation statistic DOT, and to compare it to the statistic TQ. We also included a recently proposed method "ACAT" by Liu and colleagues [18], where association P-values for individual SNPs are transformed to Cauchy-distributed random variables, then added up to obtain the overall P-value. ACAT was included into comparisons because it has robust power across different models of association. Specifically, Liu et al. found ACAT to be competitive against popular methods, including SKAT and burden tests for rare-variant associations [19–22]. A distinctive feature of ACAT is its good type-I error control in the presence of correlation between P-values, which, interestingly, improves as the $\alpha$-level becomes smaller, due to its usage of transformation to a moment-free Cauchy distribution. Among other similar approaches is MAGMA [23]. MAGMA analyzes summary association statistics by considering the mean of the chi-square statistic for the SNPs in a gene or the largest statistic among the SNPs in a gene. The mean of statistics method is equivalent to Fisher's method for combining dependent P-values [24, 25]. The method based on the top chi-square statistic among the SNPs in a gene is equivalent to the Bonferroni correction for dependent tests. There have been extensive studies comparing these two methods [26]. Note that TQ is very similar to the Fisher method.

We used two distinct scenarios in our simulation experiments:

1. First, we assumed that the summary statistics and the sample correlation matrix among statistics are estimated from the same data set. This allowed us to validate power properties derived in "Materials and Methods."

2. Second, we assumed that the sample correlation LD matrix was obtained from external reference panel. We included this scenario into our simulations due to the concern that the type-I error rate of the methods considered here may be inflated if the correlation matrix is computed based on a separate data set.

## Simulations assuming that the LD matrix and the summary statistics are obtained from the same data

To compare methods with and without decorrelation of statistics, we considered several distinct settings. In settings 1-4, the results of each row of the tables were based on one million simulations. Association statistics were simulated directly, namely, a $10^6$ by $L$ matrix of MVN vectors was simulated first, and then each row of the matrix was analyzed by the competing methods. The empirical powers were obtained as the proportion of times that a particular statistic value exceeded $\alpha = 0.05$.

Setting 1. The decorrelation method (DOT) is expected to gain power as the number of SNPs increases in scenarios where effect sizes vary markedly from SNP to SNP. However, if effect sizes for all SNPs are in fact very close to each other, the power of DOT may decrease. To illustrate this property, our first, and purposely contrived simulation setup is where the induced effect sizes (mean values of statistics) were all non-zero but very close to each other in their magnitude, varying uniformly from 2.3 to 2.4 (these are the values of the means of normally distributed standardized statistics). Table 1 shows the results of the simulations study under this setting, in which the decorrelation method was deliberately set up to fail. In the table, the columns labeled "Theoretic." provide power calculated based on the distribution of the test statistics under the alternative hypothesis that we derived above. The columns labeled "Empiric." provide results based on the empirical evaluation of power by computing P-values under the null. The columns labeled "Approx." provide power calculated based on the Eq (17). The column labeled $\bar{\gamma}$ provides the average noncentrality value.

The table illustrates that our analytical calculations under the alternative hypothesis are correct. That is, the empirical power of both TQ and DOT statistics matches nearly exactly the analytical calculations. The approximation based on Eq (17) apparently works well as well, emphasizing the fact that the distribution of the TQ statistic can be well approximated by a one-degree of freedom chi-square distribution.

**Table 1. Power comparison of TQ, DOT, and ACAT, assuming very similar effect sizes in magnitude and equicorrelation LD structure with $\rho$ = 0.7.**

| Number of SNPs | Empiric. | Theor. | Approx. | Empiric. | Theor. | ACAT | $\bar{\gamma}$ |
|---|---|---|---|---|---|---|---|
| $L$ | TQ | TQ | TQ | DOT | DOT | | |
| 500 | 0.802 | 0.802 | 0.802 | 0.090 | 0.090 | 0.832 | 0.02 |
| 300 | 0.801 | 0.801 | 0.801 | 0.101 | 0.100 | 0.830 | 0.03 |
| 200 | 0.801 | 0.801 | 0.801 | 0.112 | 0.112 | 0.829 | 0.04 |
| 100 | 0.799 | 0.800 | 0.800 | 0.144 | 0.145 | 0.826 | 0.08 |
| 50 | 0.798 | 0.799 | 0.799 | 0.196 | 0.197 | 0.821 | 0.16 |
| 30 | 0.795 | 0.796 | 0.796 | 0.253 | 0.252 | 0.814 | 0.26 |
| 20 | 0.794 | 0.793 | 0.794 | 0.307 | 0.306 | 0.809 | 0.39 |

Further, the table confirms that the decorrelation method is under-performing relative to TQ if there is very little heterogeneity among effect sizes. However, power of all methods would increase under lower correlation. For example, for $\rho = 0.3$ and $L = 20$, the powers for TQ and DOT become 0.98 and 0.67, respectively. Additional insight into power behavior of methods under this scenario can be gained by examining Eq (19). The asymptotic power for TQ can be simply computed in R as `1-pchisq(qchisq(1-0.05, df = 1), df = 1, ncp = 2.35^2/0.7)`. This gives 0.802 TQ power as $L \to \infty$ for Table 1 and 0.99 for the situation when $\rho$ is lowered to 0.3. This simple approximation is surprisingly precise and works well for the rest of the settings. Scenario 1 is admittedly unrealistic in practice. Furthermore, the table also illustrates that as the average non-centrality value increases, the power of DOT increases as well, while the power of TQ is relatively constant and about 80%. Finally, Table 1 shows that the power of TQ (although higher than that of DOT) does not change with $L$, highlighting the ceiling property of this method and the fact that combining more SNPs would not lead to higher power of TQ.

Setting 2.  One of the features of the decorrelation method is that it benefits from heterogeneity in pairwise LD. To illustrate this property, we added jiggle to the equicorrelation matrix as described in the "Materials and Methods" section, while keeping the effect size (mean values of statistics) vector the same as in Setting 1 (within the range of 2.3 to 2.4). Again, effect sizes were all non-zero. In this second set of simulations, uniformly distributed perturbations (in the range 0 to 5) were added through **U**, which made the pairwise correlations range from 0.14 to 0.98.

Table 2 summarizes the results and once again, illustrates the ceiling feature of TQ power. However, the power of the statistic DOT now starts to climb up with $L$ and the proposed test based on DOT eventually becomes more powerful than the one based on TQ. This phenomenon can be explained by examining the eigenvectors of the correlation matrix in Scenario 1. When eigenvectors are written in the form of the Helmert eigenvectors, the first contributing DOT statistic is formed as the mean of original (non-transformed) statistics. The rest of contributing statistics are weighted sums of the original statistics with weights given by the entries of $(2, \ldots, L)$ Helmert eigenvectors. However, the structure of each vector is such that its entries add up to zero (and may contain zeros as well). Thus, when the means are very similar (as in Scenario 1), there is cancellation of individual terms when the sum is formed. Moreover, note that although the

**Table 2. Power comparison of TQ, DOT, and ACAT, assuming very similar effect sizes but heterogeneous LD structure.**

| Number of SNPs | Empiric. | Theor. | Approx. | Empiric. | Theor. | ACAT | $\bar{\gamma}$ |
|---|---|---|---|---|---|---|---|
| $L$ | TQ | TQ | TQ | DOT | DOT | | |
| 500 | 0.729 | 0.730 | 0.726 | 0.973 | 0.973 | 0.793 | 0.251 |
| 300 | 0.731 | 0.730 | 0.726 | 0.883 | 0.883 | 0.791 | 0.256 |
| 200 | 0.731 | 0.730 | 0.726 | 0.810 | 0.811 | 0.789 | 0.281 |
| 100 | 0.730 | 0.731 | 0.726 | 0.599 | 0.599 | 0.786 | 0.295 |
| 50 | 0.732 | 0.733 | 0.728 | 0.577 | 0.576 | 0.782 | 0.418 |
| 30 | 0.736 | 0.735 | 0.729 | 0.504 | 0.502 | 0.778 | 0.488 |
| 20 | 0.737 | 0.737 | 0.731 | 0.541 | 0.540 | 0.776 | 0.661 |

**Table 3. Power comparison of TQ, DOT, and ACAT, assuming heterogeneity in effect sizes but equicorrelated LD.**

| Number of SNPs | Empiric. | Theor. | P-approx. | Empiric. | Theor. | ACAT | $\bar{\gamma}$ |
|---|---|---|---|---|---|---|---|
| $L$ | TQ | TQ | TQ | DOT | DOT | | |
| 500 | 0.525 | 0.525 | 0.526 | 1.000 | 1.000 | 0.626 | 0.479 |
| 300 | 0.526 | 0.525 | 0.526 | 1.000 | 0.999 | 0.624 | 0.486 |
| 200 | 0.526 | 0.525 | 0.524 | 0.993 | 0.993 | 0.622 | 0.494 |
| 100 | 0.525 | 0.524 | 0.524 | 0.919 | 0.920 | 0.616 | 0.518 |
| 50 | 0.522 | 0.523 | 0.522 | 0.762 | 0.762 | 0.607 | 0.566 |
| 30 | 0.521 | 0.521 | 0.521 | 0.648 | 0.648 | 0.599 | 0.630 |
| 20 | 0.519 | 0.519 | 0.520 | 0.578 | 0.579 | 0.592 | 0.709 |

average noncentrality value does not increase with $L$, the DOT-test still gains power with $L$!

Setting 3. This setting is analogous to the equicorrelation scenario in Setting 1, except that the mean values of statistics were lowered: in Setting 1, the range in $\mu$ was 2.3 to 2.4, while here, the range was set to vary uniformly between 1 and 2.3, and effect sizes were all non-zero. Thus, the maximum effect size was lower than that in the previous simulations but the heterogeneity among effect sizes was higher. We emphasize again that while the equicorrelation assumption is unrealistic, it serves as a very useful benchmark scenario that highlights power behavior and features of the statistics TQ and DOT and allows one to introduce departures from equicorrelation in a controlled manner.

Table 3 presents the results. The "Approx." column in this table was removed and replaced by power values based on a "P-value"-approximation to the distribution of TQ as in Eq (16). This switch highlights the idea that both the power and the P-value for the TQ test can be reliably estimated based on the one degree of freedom chi-squared approximation. Importantly, Table 3 demonstrates that the power of the DOT-test reaches 100% as $L$ increases (despite the fact that effect sizes were lower than in the previous settings), while the power of the TQ-test stays in the range 51.2 to 52.5%.

Setting 4. This setting is similar to the scenario in Setting 2, except that we allowed higher heterogeneity in pair-wise LD values. Effect sizes were all non-zero. LD was constructed as perturbation of $\mathcal{R}_{\rho=0.7} + \mathbf{U}\mathbf{U}'$ (as described in "Materials and Methods"), with $\mathbf{U}$ set to be a random sequence on the interval from -5 to 5. This resulted in LD values ranging from -0.93 to 0.99. The effect sizes (mean values of statistics) were sampled randomly within each simulation from (-0.15, 0.15) interval.

Table 4 presents the results and shows that in this setting, the power of DOT is dramatically higher than that of TQ and ACAT. In fact, power values for the TQ and ACAT tests barely exceed the type-I error, while the power of the decorrelation method steadily increases with $L$, eventually exceeding 90%.

Settings 5–7. In these sets of simulations we used biologically realistic patterns of LD. Also, rather than specifying mean values of association statistics directly, we utilized a regression model for the effect sizes, as described in Eqs (1) and (2). Details of these simulations are given in "LD patterns from the 1000 Genome Project" in "Materials and Methods." We re-iterate that when association of SNPs with a

**Table 4. Power comparison of TQ, DOT, and ACAT with effect sizes randomly sampled from -0.15 to 0.15 and heterogeneous LD.**

| Number of SNPs | Empiric. | Theor. | P-approx. | Empiric. | Theor. | ACAT | $\bar{\gamma}$ |
|---|---|---|---|---|---|---|---|
| $L$ | TQ | TQ | TQ | DOT | DOT | | |
| 500 | 0.0500 | 0.0503 | 0.0508 | 0.9226 | 0.9222 | 0.0564 | 0.2118 |
| 300 | 0.0506 | 0.0503 | 0.0509 | 0.7688 | 0.7689 | 0.0570 | 0.2107 |
| 200 | 0.0504 | 0.0503 | 0.0508 | 0.5970 | 0.5967 | 0.0570 | 0.2025 |
| 100 | 0.0504 | 0.0503 | 0.0509 | 0.3040 | 0.3038 | 0.0568 | 0.1655 |
| 50 | 0.0502 | 0.0503 | 0.0508 | 0.3074 | 0.3070 | 0.0555 | 0.2397 |
| 30 | 0.0505 | 0.0503 | 0.0507 | 0.1485 | 0.1487 | 0.0562 | 0.1527 |
| 20 | 0.0501 | 0.0503 | 0.0508 | 0.1191 | 0.1189 | 0.0557 | 0.1399 |

trait is present (under the alternative hypothesis), the correlation among statistics is not equal to LD, because it also has to incorporate effect sizes, as illustrated by Eq (5). This point is important if one wants to simulate statistics directly from the MVN distribution rather than computing them based on simulated data followed by regression.

The results are presented in Table 5. Columns labeled "Regr." represent scenarios, in which data were generated and statistics were computed. Columns labeled "MVN" represent scenarios, in which statistics were simulated directly. The rows of Table 5 show power values for three different $\alpha$-levels. We expected the power values in "Regr." and "MVN" columns to match, and they do, highlighting another utility of our analytical derivation of the distribution of the test statistic under the alternative hypothesis. That is, using our results, one can significantly reduce computational and programming burden in genetic simulations. Also note that power values in Table 5 do not decrease as $\alpha$-level becomes smaller (Settings 6 and 7). This is due to the fact that we deliberately discarded effect size and LD configurations where power was expected to be too low, because we wanted to assure a good range of power values across methods.

As in previous simulations, power values of TQ and ACAT are similar. The power approximation by Eq (17) remains close to the predicted theoretical power, as well as to empirically estimated powers. We also observed that power of the decorrelation test, DOT, is substantially higher than the powers of either TQ or ACAT.

Patterns of LD and effect sizes in Settings 1–4 are not necessarily realistic biologically, however, they serve as benchmark scenarios that help to understand and highlight differences in the respective statistical power of the methods. Simulations for Settings 1–4 were performed at

**Table 5. Power comparison of TQ, DOT, and ACAT using realistic LD patterns from 1000 Genomes project.**

| | Theor. | Approx. | Regr. | MVN | Theor. | Regr. | MVN | |
|---|---|---|---|---|---|---|---|---|
| | TQ | TQ | TQ | TQ | DOT | DOT | DOT | ACAT |
| Setting 5 | | | | | | | | |
| $\alpha = 10^{-3}$ | 0.34 | 0.34 | 0.34 | 0.34 | 0.60 | 0.60 | 0.60 | 0.40 |
| Setting 6 | | | | | | | | |
| $\alpha = 10^{-4}$ | 0.42 | 0.42 | 0.42 | 0.43 | 0.77 | 0.77 | 0.77 | 0.43 |
| Setting 7 | | | | | | | | |
| $\alpha = 10^{-7}$ | 0.24 | 0.24 | 0.24 | 0.24 | 0.76 | 0.76 | 0.76 | 0.18 |

**Table 6. Type-I error rates ($\alpha = 10^{-3}$) using a reference panel to estimate LD.** Population LD patterns are modeled using 1000 Genomes project data.

| Sample size | TQ | DOT | ACAT |
|:---:|:---:|:---:|:---:|
| $N = 5L$ | $1 \times 10^{-3}$ | $3 \times 10^{-3}$ | $1 \times 10^{-3}$ |
| $N = 10L$ | $1 \times 10^{-3}$ | $3 \times 10^{-3}$ | $1 \times 10^{-3}$ |
| $N = 50L$ | $1 \times 10^{-3}$ | $2 \times 10^{-3}$ | $1 \times 10^{-3}$ |
| $N = 100L$ | $1 \times 10^{-3}$ | $1 \times 10^{-4}$ | $1 \times 10^{-3}$ |

the 5% $\alpha$-level based on $2 \times 10^6$ evaluations. Settings 5–7 used realistic patters of LD derived from the 1000 Genomes Project data. Test sizes varied from 0.001 to $10^{-7}$ with at least 10,000 simulations for power estimates. Type-I error rates were well controlled for TQ and DOT. However, as noted by Liu et al., because the ACAT P-value is approximate, the null distribution of its statistic is evaluated under independence, and we found that at the nominal 5% $\alpha$-level, the type-I error for the ACAT was somewhat higher and could reach 7% for some correlation settings. Nonetheless, the advantage of ACAT is that the approximation improves as the $\alpha$-level becomes smaller.

## Simulations assuming that the correlation matrix is estimated using external data

When only summary statistics are available, the correlation matrix $\Sigma$ can be estimated from a reference panel of genotyped individuals. However, the type-I error of tests based on both TQ and DOT may potentially be affected due to substituting the sample estimate $\hat{\Sigma}$ by an estimate obtained from external data. To study the effect of this mis-specification on the type-I error, we conducted a separate set of simulations. In these experiments, we again utilized LD structures derived from the 1000 Genomes Project data. Reference panels for these simulations were obtained as follows. Each LD matrix derived from real data was assumed to represent the population matrix. Next, a sample was drawn, and the corresponding sample LD matrix was calculated. That matrix should have been used for calculations of the gene-based test statistics. Instead, we drew a separate sample of size $N$, assuming the same population LD matrix. In the calculation of the tests, that sample correlation matrix was used in place of the correct one. The type-I error rates, given in Tables 6–8, show that both ACAT and TQ have close to the nominal type-I error rates, but the error rate for the decorrelation method (DOT) can be inflated, unless the sample size of the reference panel is 50 to 100 times larger than the number of SNPs ($L$). For the statistic DOT, the type-I error rates appear to be more inflated at smaller $\alpha$-levels, such as $10^{-7}$. Power values for TQ are not shown, however they closely followed predicted theoretical power for the scenarios where the same data are used for both LD estimation

**Table 7. Type-I error rates ($\alpha = 10^{-4}$) using a reference panel to estimate LD.** Population LD patterns are modeled using 1000 Genomes project data.

| Sample size | TQ | DOT | ACAT |
|:---:|:---:|:---:|:---:|
| $N = 5L$ | $9 \times 10^{-5}$ | $5 \times 10^{-4}$ | $1 \times 10^{-4}$ |
| $N = 10L$ | $9 \times 10^{-5}$ | $4 \times 10^{-4}$ | $1 \times 10^{-4}$ |
| $N = 50L$ | $1 \times 10^{-4}$ | $1 \times 10^{-4}$ | $1 \times 10^{-4}$ |
| $N = 100L$ | $1 \times 10^{-4}$ | $1 \times 10^{-4}$ | $1 \times 10^{-4}$ |

**Table 8. Type-I error rates ($\alpha = 10^{-7}$) using a reference panel to estimate LD.** Population LD patterns are modeled using 1000 Genomes project data.

| Sample size | TQ | DOT | ACAT |
|:---:|:---:|:---:|:---:|
| $N = 5L$ | $2 \times 10^{-7}$ | $3 \times 10^{-4}$ | $1 \times 10^{-7}$ |
| $N = 10L$ | $2 \times 10^{-7}$ | $2 \times 10^{-4}$ | $1 \times 10^{-7}$ |
| $N = 50L$ | $2 \times 10^{-7}$ | $2 \times 10^{-4}$ | $1 \times 10^{-7}$ |
| $N = 100L$ | $2 \times 10^{-7}$ | $1 \times 10^{-4}$ | $1 \times 10^{-7}$ |

and computation of association statistics. There was only 1 to 2% drop in power when the size of the panel was only 2 to 5 times larger than *L*.

## Combining breast cancer association statistics within candidate genes

We applied our decorrelation method to a family-based GWAS study of breast cancer [27, 28]. The data set was comprised of complete trios, i.e., families where genotypes of both parents and the affected offspring were available. With complete trios, previously reported statistics become equivalent to statistics from the transmission-disequilibrium test and correlation among them is expected to follow the LD among SNPs [8]. We selected four candidate genes (*TOX3, ESR1, FGFR2* and *RAD51B*), for which Shi et al. [27] and O'Brien et al. [28] replicated several previously reported risk SNPs in relation to breast cancer.

For the joint association, we restricted our analysis to blocks of SNPs surrounding breast cancer risk variants that were previously reported in the literature. Specifically, we selected *TOX3* rs4784220 [29], *ESR1* rs3020314 [30, 31], *FGFR2* rs2981579 [29], and *RAD51B* rs999737 [32–34], and then included blocks of SNPs around these 'anchor' risk variants with the LD correlation of at least 0.25. These blocks included 13 SNPs around rs4784220, 36 SNPs around rs3020314, 18 SNPs around rs2981579, and 30 SNPs around rs999737. As an illustration, Fig 1 displays 81 SNP P-values that were available for *ESR1* gene, the vertical dashed line highlights the position of 'anchor' rs3020314, the red dots highlight 36 SNPs within LD-block of rs3020314, and the LD matrix displays sample correlation matrix among 36 SNPs. Once SNP blocks were identified for each gene, we applied four combination methods to assess their association with breast cancer.

Table 9 present the joint association analysis results. The first row of Table 9 shows P-values for the association between the LD block of 13 SNPs in *TOX3* region and breast cancer, derived from 1277 Caucasian triads. All methods conclude a statistically significant link but our decorrelation method provides the most robust evidence with a substantially lower P-value. The third row of Table 9 shows joint association P-values for the LD block of 18 SNPs in *FGFR2*. Three out of four methods conclude an association at 5% level, with DOT approach, once again, providing the most significant result. We note that the last column of Table 9 gives the Bonferroni-style adjustment that is expected to be more conservative relative to the combination tests. Thus, it is not surprising that out of the four methods considered, the Bonferroni method failed to conclude an association. Lastly, the second and the fourth rows of Table 9 provide joint association P-values for LD block in *ESR1* and *RAD51B*, respectively. For both *ESR1* and *RAD51B* our decorrelation approach was the only one that concluded a statistically significant association between SNP-set in those genes with breast cancer.

Table 10 details a list of top SNPs that are associated with breast cancer within the selected candidate genes. The top ranked SNPs were identified by considering the top three components in the linear combination $\text{DOT} = \sum_{i=1}^{L} X_i^2$, where $X_i$'s are the decorrelated summary

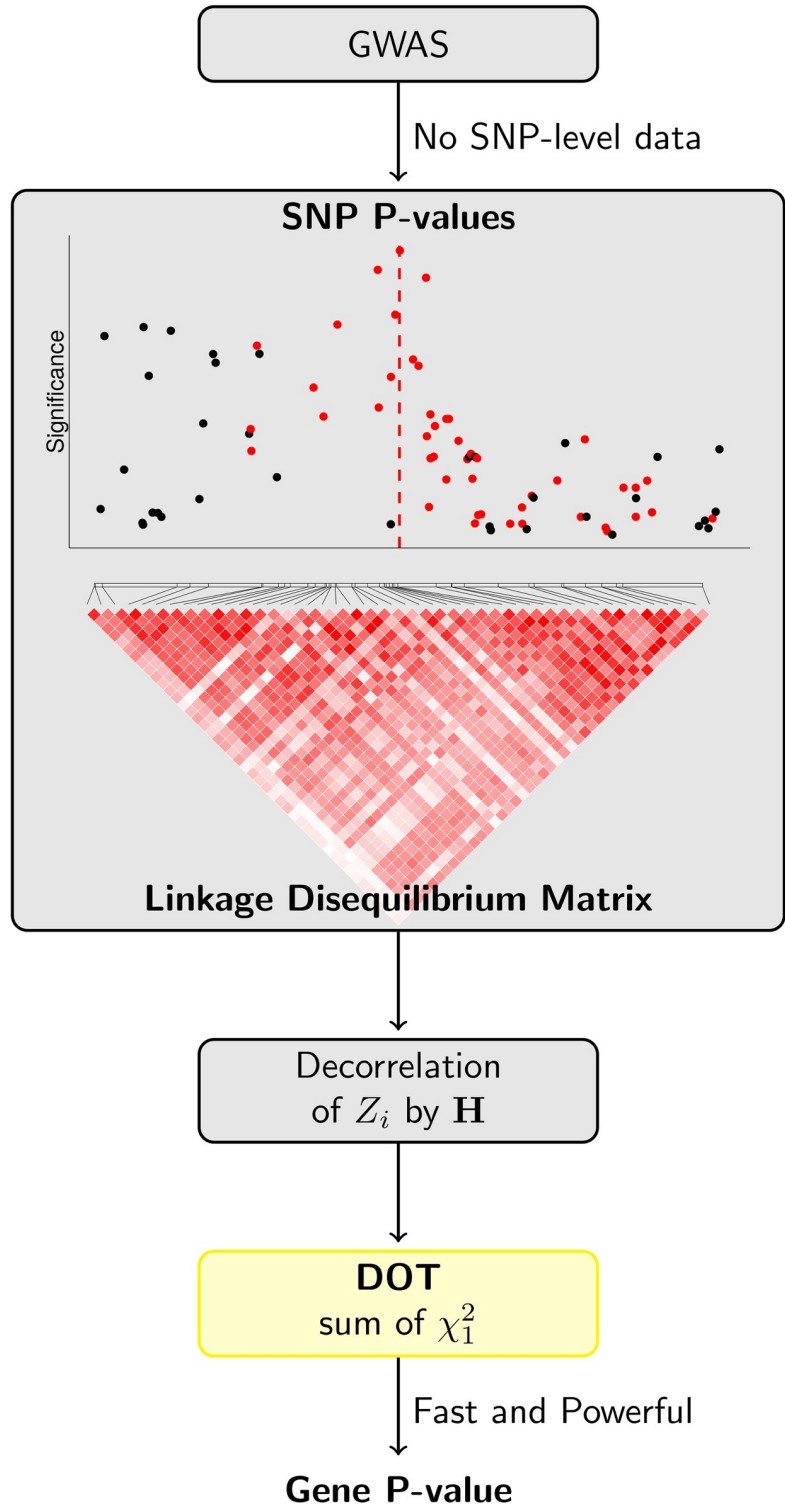

**Fig 1. Overview of DOT method in application to breast cancer data.** We compute gene-level score by first decorrelating SNP P-values using the invariant to order matrix **H** and then calculating sum of independent chi-squared statistics. We utilize our DOT method to obtain a gene-level P-value. In the breast cancer data application, we chose an anchor SNP—a SNP that has previously been reported as risk variant (highlighted by a vertical dashed line), —and then combine SNPs in an LD block with the anchor SNP by the DOT. SNP-level P-values highlighted in red are those in moderate to high LD with the anchor SNP.

**Table 9. Breast cancer candidate gene association P-values.**

| Gene | TQ | DOT | ACAT | $\min(P) \times L$ |
|------|-----|------|------|------|
| *TOX3*/rs4784220 [29] ($L = 13$) | 0.0005 | 0.0004 | 0.001 | 0.001 |
| *ESR1*/rs3020314 [30, 31] ($L = 36$) | 0.20 | 0.0001 | 0.19 | 0.96 |
| *FGFR2*/rs2981579 [29] ($L = 18$) | 0.01 | 0.003 | 0.01 | 0.07 |
| *RAD51B*/rs999737 [32–34] ($L = 30$) | 0.56 | 0.009 | 0.76 | 1 |

statistics. Once the highest three values of $X_i^2$ were identified for each gene, we considered individual components of $X_i = \sum_{j=1}^{L} h_j Z_j$ that are formed as a linear combination of the original statistics weighted by the elements of matrix **H**. The top individual components $h_j Z_j$ (with the same sign as $X_i$) were corresponding to individual SNPs presented in Table 10.

For the LD block in *TOX3* gene, the top three individual $X_i$'s in DOT statistic were all formed by having a very large weight assigned to a single SNP, i.e., the largest value, $X_{(1)}^2$, was formed by assigning a large weight to rs4784220 statistic; the second largest value, $X_{(2)}^2$, was formed by assigning a large weight to rs8046979 statistic; and the third largest value, $X_{(3)}^2$, was formed by assigning a large weight to rs43143 statistic. The first few rows of Table 10 detail these results and identify rs43143 as a new possible association with breast cancer.

For the LD block in *ESR1* gene, the top $X_i$'s were quite different. Specifically, the largest value, $X_{(1)}$, was formed as a linear combination of 6 SNPs that all got assigned large weights. These 6 SNPs were rs2982689/rs3020424/rs985695/rs2347867/rs3003921/rs985191. The second highest linear combination, $X_{(2)}$, was formed by assigning high weights to 5 out of 6 SNPs

**Table 10. Breast cancer SNPs identified by DOT in the analysis of GWAS data.**

| Gene | Number of SNPs in analysis ($L$) | rs number | Reference |
|------|------|------|------|
| *TOX3* | 13 | rs4784220 | This SNP was previously reported in the literature to be associated with breast cancer [29, 35]. |
| | | rs8046979 | This SNP was also linked to breast cancer [29]. |
| | | **rs43143** | A new association with susceptibility to breast cancer. |
| *ESR1* | 36 | rs2347867 | This SNP was previously reported to be involved in breast cancer risk [36, 37]. |
| | | rs985191 | This SNP was previously reported to be associated with endocrine therapy efficacy in breast cancer [38], as well as with the overall breast cancer risk [39]. |
| | | **rs3003921** | A new association with susceptibility to breast cancer. This SNP was previously linked to the effectiveness of androgen deprivation therapy among prostate cancer patients [40]. |
| | | **rs985695** | A new association with susceptibility to breast cancer. |
| | | **rs2982689** | A new association with susceptibility to breast cancer. |
| | | **rs3020424** | A new association with susceptibility to breast cancer. |
| | | **rs926777** | A new association with susceptibility to breast cancer. |
| *FGFR2* | 18 | rs1219648 | This SNP was previously reported to be associated with premenopausal breast cancer [41] and the overall breast cancer risk [42–45]. |
| | | rs2860197 | This SNP was previously suggested to have an association with breast cancer [46]. |
| | | rs2981582 | This SNP was previously reported in the literature to be associated with breast cancer [43, 47–49]. |
| | | rs3135730 | This SNP was previously suggested to have an interaction between oral contraceptive use and breast cancer [50]. |
| | | **rs2981427** | A new association with susceptibility to breast cancer. |
| *RAD51B* | 30 | rs999737 | This SNP was previously reported in the literature to be associated with breast cancer [32–34, 51, 52]. |
| | | rs8016149 | This SNP was previously suggested to have an association with breast cancer [53]. |
| | | rs1023529 | This SNP has been patented as one of susceptibility variants of breast cancer [54]. |
| | | rs2189517 | This SNP was showed to be associated with breast cancer in Chinese population [55]. |
| | | **rs7359088** | A new association with susceptibility to breast cancer. |

listed above: rs2982689/rs3020424/rs985695/rs2347867/rs3003921. We note that the signs of $X_{(1)}$ and $X_{(2)}$ were in different directions and that is why it was possible for the same set of SNPs to be prioritized. Finally, the third largest value, $X_{(3)}$, also prioritized the same set of SNPs, with the exception of the single new addition of rs926777. Table 10 provides a detailed discussion of these SNPs and identifies rs3003921/rs985695/rs2982689/rs3020424 and rs926777 as new possible associations with breast cancer.

Finally, for the LD blocks in *FGFR2* and *RAD51B* we repeated the procedure detailed above and also identified top-ranking SNPs. Table 10 reviews these results and points *FGFR2* rs2981427 and *RAD51B* rs7359088 as two more additional newly found associations.

## Combining cleft lip association statistics within candidate genes

To further validate the utility of DOT, we applied it to summary statistics of a recent GWAS of cleft lip with and without cleft palate [56]. Summary statistics were based on transmission-dis-equilibrium test on autosomal SNPs in 1908 case-parent trios of European and Asian ancestry. We selected four genetic regions (*ABCA4*, chr. 8q24, *IRF6*, and *MAFB*) that were prioritized by Beaty et al. [56] for gene-based analysis. Anchor SNPs were chosen based on significant risk markers previously reported in literature. Specifically, rs560426 was chosen as an anchor for *ABCA4* region [57] and formed an anchor block of $L = 30$ SNPs; rs987525 for chr. 8q24 [58] with $L = 29$ SNPs in a block; rs10863790 for *IRF6* [59] with $L = 6$ SNPs in a block; and rs13041247 for *MAFB* [60] with $L = 14$ SNPs in a block. Table 11 provides summary of gene-based P-values and indicates that all four combination methods concluded significant associations. Results in Table 11 can also be viewed as a gauge of the relative power of the four combination methods. As such, Table 11 confirms that DOT may result in smaller P-values then those of competitors.

Table 12 details a list of top SNPs that were associated with non-syndromic cleft lip with or without cleft palate within four genetic regions. For the LD block around rs560426 in *ABCA4* gene, $X_{(1)}^2$ was formed by assigning large weights to two SNPs (rs4847196/rs563429) both of which were previously considered in association with cleft lip but were found to be not statistically significant [56]. The second highest DOT linear combination, $X_{(2)}^2$, prioritized the same two SNPs (rs4847196/rs563429), thus reinforcing the idea that these two markers may be genuinely associated with cleft lip. The third highest linear combination, $X_{(3)}^2$, was formed by assigning high weights to rs2275035 and rs546550, the former of which was recently identified to be associated with orofacial clefting [61], while the latter may be a new association with cleft lip.

For the LD block on chr. 8q24 region, $X_{(1)}^2$ was formed by assigning a large weight to the anchor SNP (rs987525). $X_{(2)}^2$ prioritize two SNPs: rs882083 that was already suggested to be associated with cleft lip [56, 58], and rs12547241 that may be a new risk marker. Finally, $X_{(3)}^2$ prioritized a set of three SNPs (rs1157136/rs12548036/rs1530300), all of which were previously studied in connection to cleft lip [57, 63–65]. For the last two LD block considered (*IRF6* and

**Table 11. Cleft lip candidate gene association P-values.**

| Gene | TQ | DOT | ACAT | min(*P*) × *L* |
|---|---|---|---|---|
| *ABCA4*/rs560426 [57] (*L* = 30) | $8.9 \times 10^{-8}$ | $1.3 \times 10^{-13}$ | $7.2 \times 10^{-11}$ | $7.2 \times 10^{-11}$ |
| chr. 8q24/rs987525 [58] (*L* = 29) | $1.0 \times 10^{-9}$ | $8.7 \times 10^{-22}$ | $4.7 \times 10^{-15}$ | $3.2 \times 10^{-15}$ |
| *IRF6*/rs10863790 [59] (*L* = 6) | $4.7 \times 10^{-9}$ | $1.8 \times 10^{-19}$ | $2.1 \times 10^{-14}$ | $2.1 \times 10^{-14}$ |
| *MAFB*/rs13041247 [60] (*L* = 14) | $1.5 \times 10^{-8}$ | $2.9 \times 10^{-8}$ | $2.4 \times 10^{-11}$ | $3.6 \times 10^{-11}$ |

**Table 12. Cleft SNPs identified by DOT in the analysis of GWAS data.**

| Gene | Number of SNPs in analysis (L) | rs number | Reference |
|---|---|---|---|
| ABCA4 | 30 | rs4847196 | This SNP was previously studied in connection to cleft lip [56] but the association was found to be not statistically significant. |
| | | rs563429 | This SNP was also previously considered in association with cleft lip [56] but found to be not statistically significant. |
| | | rs2275035 | Was recently identified to be associated with orofacial clefting [61]. |
| | | **rs546550** | A new association with susceptibility to cleft lip. This SNP was previously suggested to be linked to esophageal cancer [62]. |
| chr. 8q24 | 29 | rs987525 | One of the top results was the anchor SNP [58]. |
| | | rs882083 | Was previously suggested to be associated with cleft lip [56, 58]. |
| | | rs1157136 | Was previously suggested to be associated with cleft lip in Brazilian population [63]. |
| | | rs12548036 | Was previously studied in connection to susceptibility to cleft lip in Japanese population [64] but the association was found to be not statistically significant. |
| | | rs1530300 | Was previously suggested to be associated with cleft lip in Brazilian population [57] and Brazilian population with high African ancestry [65]. |
| | | **rs12547241** | A new association with susceptibility to cleft lip. |
| IRF6 | 6 | rs10863790 | One of the top contributions was the anchor SNP [59]. |
| | | rs861020 | Was previously reported to be associated with cleft lip [59, 66, 67]. |
| | | rs2236906 | Was considered to be associated with cleft lip in a Kenya African Cohort [68] and in general population [69]. |
| | | rs2073485 | Was reported to be associated with cleft lip in Western China [70] and Taiwanese population [71]. |
| MAFB | 14 | rs11696257 | Was previously reported to be associated with cleft lip [56, 72]. |
| | | rs6102085 | Was previously reported to be associated with cleft lip in Han Chinese population [73]. |
| | | rs6065259 | Was previously reported to be associated with cleft lip in a population in Heilongjiang Province, northern China [74]. |
| | | rs6102074 | Was previously reported to be associated with cleft lip in Han Chinese population [73, 75]. |

*MAFB* genes), Table 12 details a list of top SNPs contributors to the DOT statistic. In brief, all of the prioritized SNPs were previously reported in association with cleft lip.

## Discussion

In this research, we have proposed a new powerful decorrelation-based approach (DOT) for combining SNP-level summary statistics (or, equivalently, P-values) and derived its theoretical power properties. To the best our knowledge, we were the first to derive analytical properties of the traditional approach, TQ (e.g., as implemented in VEGAS), as well as of the DOT, with the help of new theory that incorporates effect sizes of SNPs into mean values of association statistics and correlations among them. Through extensive simulation studies, we have demonstrated that our decorrelation approach is a powerful addition to the tools available for studying genetic susceptibility to disease.

Our analysis of breast cancer and cleft lip data illustrates unique properties of DOT. Our results revealed novel potential associations within candidate genes that would have not been found by previously proposed methods. These novel SNPs were identified by examining the top three linear-combination contributors to the overall value of the DOT-statistic. We note that the top contributions may give large weights to genetic variants that are truly associated with the outcome or to SNPs in a high positive LD with true causal variants. Caution is needed when interpreting such results because our method cannot distinguish between causal and proxy associations. Further studies would be needed to confirm these findings.

The most important feature of the proposed method is that it may provide substantial power boost across diverse settings, where power gain is amplified by heterogeneity of effect sizes and by increased diversity between pairwise LD values. Genetic architecture of complex traits is far from being homogeneous, making our method applicable in various settings. We have developed new theory to explain unexpected and remarkable boost in power. This theory allows one to predict behavior of the tests in simulations with high accuracy and to explain unexpected scenarios, where the decorrelation method may give dramatically higher power compared to the traditional approach. Yet, there are important precautions to the decorrelation approach. When reference panel data are used to provide the LD information and, more generally, correlation estimates for all predictors, including SNPs and covariates, $\hat{\Sigma}$, sample size of the external data should be several times larger than the number of predictors. Ideally, the same data set should be used to obtain association statistics, as well as $\hat{\Sigma}$. Nevertheless, association statistics and $\hat{\Sigma}$ are compact summaries of data and are much more easily transferred between separate research groups than raw data, due to privacy considerations and potentially large size of the raw data sets. Also, caution is needed if missing data are present in the original data set because the estimate ($\hat{\Sigma}$) may no longer reflect the sample correlation between predictors. Imputation of missing values is a suitable solution, if missing values are independent of the outcome. With the usage of reference panel data, the type-I error inflation for the statistic DOT can be affected by many factors, and this statistic is expected to be sensitive not only to the size of a reference panel, but to population variations in LD, especially for highly correlated blocks of SNPs. Overall, it appears to be difficult to give specific recommendations, except that the reference panel size has to be at least 50 times larger than the number of SNPs to be combined. Therefore, we recommend to limit applications of the decorrelation method to situations, where the LD matrix is obtained from the same data set as the summary statistics. Note that all pairwise LD values can be obtained from sample haplotype frequencies of SNPs, thus the LD matrix can be reconstructed. Utility of this approach remains to be investigated, in particular, one concern is that the correlation between the SNP values reflect the composite disequilibrium values [76], while frequencies of sample haplotypes are often reported following likelihood maximization, e.g., by the EM algorithm. An important issue that still remains to be investigated is a systematic analysis of the performance of our method utilizing real genome-wide data. Such analysis would allow one a more thorough assessment of both the type-I error rate, as well as power to detect genetic regions already implicated in susceptibility to disease.

In our simulations, the recently proposed method ACAT and the test based on the distribution of the sum of correlated association statistics (VEGAS, or TQ) had similar power. In many situations, power of these two tests was substantially lower than that of the DOT. The main advantage of ACAT is that it does not require any LD information. Our theory and simulations also revealed previously unknown robustness of the TQ method with respect to LD mis-specification: the method is valid and remains nearly as powerful when the sample LD matrix is substituted by a single value, summarizing the extent of all pairwise correlations. TQ also remains valid when the LD summary is obtained from a representative reference panel. We stress again that compared to ACAT and TQ, our method's limitation is that in order to avoid possible bias, the LD information and the summary statistics should ideally come from the same data set and missing genotypes should be imputed prior to its application. In general, one should avoid utilization of external data as a source of LD information, as well as high rates of unimputed missing genotypes. Although not pursued here, a possible way to improve robustness of the DOT is to merge it with ACAT, that is, decorrelate the summary statistics first, convert the results to P-values and then combine them with ACAT.

## Materials and methods

Genetic association tests based on summary statistics are often presented as a weighted sum [2, 4]. Let $w_i$ denote the weight assigned to individual statistic. The weighted statistics can then be defined as $Y_i^2 = w_i Z_i^2$ with $\mathbf{Z} \sim \text{MVN}(\boldsymbol{\mu}_Z, \boldsymbol{\Sigma}_Z)$ and $\mathbf{Y} \sim \text{MVN}(\boldsymbol{\mu}, \boldsymbol{\Sigma})$, where $\boldsymbol{\mu} = \mathbf{W}\boldsymbol{\mu}_Z$, $\boldsymbol{\Sigma} = \mathbf{W}\boldsymbol{\Sigma}_Z\mathbf{W}$, and $\mathbf{W} = \text{diag}(\sqrt{\mathbf{w}})$. The statistics $Y_i^2$ are marginally distributed as one degree of freedom chi-square variables with noncentralities $\mu_i^2$. The overall statistic is then typically defined as $\text{TQ} = \sum_{i=1}^{L} Y_i^2$.

## Joint distribution of association summary statistics

In this section, we derive parameters $\boldsymbol{\mu}$ and $\boldsymbol{\Sigma}$ of the joint MVN distribution of summary statistics. Under the null hypothesis, when none of the SNPs are associated with an outcome, $\boldsymbol{\mu} = \mathbf{0}$. If individual SNP models do not include covariates, $\boldsymbol{\Sigma}_Z$ equals the LD matrix, i.e., the correlation matrix between the SNP values coded as 0, 1, or 2, reflecting the number of minor alleles in a genotype. In the presence of covariates, $\boldsymbol{\Sigma}_Z$ is a Schur complement of the submatrix of the matrix of all predictor variables [6]. That is, the estimated correlation between association statistics $\hat{\boldsymbol{\Sigma}}_Z$ can be obtained by inverting the covariance or correlation matrix of all predictors, selecting the SNP submatrix, inverting it back, and standardizing the result to correlation.

Under the alternative hypothesis, when some SNPs are associated with a trait $y$, let $\beta_j$ be the regression coefficient for the $j$-th SNP. Then, a typical linear model that determines the trait value is defined as:

$$y = \beta_0 + \sum_{j=1}^{L} \beta_j \,\text{SNP}_j + \epsilon, \tag{1}$$

where $\epsilon \sim N(0, 1)$. The mean value of the summary statistics (i.e., noncentralities) can be expressed as:

$$\mu_j = \sqrt{N}\frac{\Sigma_j \boldsymbol{\beta}}{\sqrt{\boldsymbol{\beta}'\Sigma\boldsymbol{\beta} + 1}} = \sqrt{N}\; b_j, \tag{2}$$

where $\Sigma_j$ is the $j$-th column of $\Sigma$, $b_j = \text{cor}(y, \text{SNP}_j)$ and $N$ is the sample size. An intuitive explanation of Eq (2) can be gained by considering the case of independent predictors, i.e., $\Sigma = \mathbf{I}_L$. If both the outcome and the set of predictors are standardized, then $\frac{\Sigma_j\boldsymbol{\beta}}{\sqrt{\boldsymbol{\beta}'\Sigma\boldsymbol{\beta}+1}} = \frac{\beta_j}{\sqrt{\sum_j \beta_j^2 + 1}}$, which is a standardized regression coefficient. We note that Eq (2) is valid outside of the linear model settings. For example, consider a latent variable model, where the continuous unobserved (latent) variable $y_l$ is linear in predictors according to Eq (1), and the observed variable (disease status) is $y = 1$ whenever $y_l > l$ and $y = 0$ otherwise, where $l$ is some threshold. When such binary outcome is analyzed by logistic regression, a good approximation to the noncentrality values will be:

$$\mu_j \approx \sqrt{N}(d \times b_j). \tag{3}$$

If error terms $\epsilon$ are assumed to be normally distributed, the reduction in correlation due to dichotomization by the factor $d$ can be expressed as $d = \phi(l)/\sqrt{\Phi(l)(1 - \Phi(l))}$, where $\phi(\cdot)$, $\Phi(\cdot)$ are the probability and the cumulative densities of the standard normal distribution [77].

Under association, surprisingly, the correlation matrix between statistics is no longer $\Sigma$. Let $\sigma_{ij}$ be the $i, j$-th element of $\Sigma$, and $\rho_{ij}$ be correlations between predicdictors and the outcome. By using the multivariate delta method, we derived the $i, j$-th element of the correlation matrix

$\mathcal{R}$ as follows:

$$\mathcal{R}_{ij} \approx \frac{(\mu_i\mu_j(\mu_i^2 + \mu_j^2 - N) - 2(\mu_i^2 + \mu_j^2 - N)N\sigma_{ij} + \mu_i\mu_j N\sigma_{ij}^2)}{(\mu_i^2 - N)(\mu_j^2 - N)},$$

$$\approx \rho_{ij} + \rho_{ij}^2 \frac{\mu_i\mu_j}{2N} - \rho_{ij}\frac{\mu_i^2\mu_j^2}{N^2} - \frac{\mu_i\mu_j}{2N}, \tag{4}$$

$$= \rho_{ij} + \rho_{ij}^2 b_i b_j - \rho_{ij}b_i^2 b_j^2 - b_i b_j. \tag{5}$$

Details of the derivation of these equations are given in [78]. An alternative derivation of the asymptotic covariance that includes the first two terms of Eq (5) has been given by Reshef et al. [79], assuming Gaussian genotypes, an assumption justifiable provided that there is a lower bound for minor allele frequency relative to sample size. Note that when some of SNP pairs $(i, j)$ are associated, summary statistics may become correlated even if there is no LD between the SNPs, due to the last term, $-b_i b_j$, in Eq (5). Eqs (2), (3), (4) and (5) allow one to study power properties of the methods based on sums of association statistics, as well as to design realistic simulation experiments, where summary statistics can be sampled directly from the MVN distribution under the alternative hypothesis. That is, given effect sizes and the correlation matrix among predictors, statistics can be immediately sampled from the MVN $(\boldsymbol{\mu}, \mathcal{R})$ distribution. This approach avoids both the data-generating step and the subsequent computation of summary statistics from that data, leading to a substantial gain in computation time. In certain situations, the difference in speed can be dramatic. For example, it is not trivial to simulate discrete (genotype) data given a specific LD matrix. Current state of the art methods tend to be slow, because they rely on ad hoc iterative techniques, such as generation of multiple random "proposal" data sets to fit the target correlation matrix [80].

Results of simulation experiments presented here were performed based on effect sizes specified via the linear model (Eq 1). However, we verified (not presented here) the validity of the proposed theory assuming logistic, probit, and Poisson regression models. We also note that Conneely et al. presented theoretical arguments supporting the validity of the MVN joint distribution of summary statistics under no association for a broad class of generalized regression models [6].

## Distribution of sums of association summary statistics

As we noted at the beginning of the "Materials and Methods" section, weighted sums of summary statistics can be re-expressed as unweighted sums, where the mean and the correlation parameters are modified to absorb the weights. The distribution of $\sum_{i=1}^{L} Y_i^2$ follows the weighted sum of independent one degree of freedom non-central chi-square random variables. Although this result is standard, the components of this weighted sum depend on the joint distribution of association summary statistics under the alternative hypothesis, and this distribution has not been previously derived. In the previous section, we provide the components of $\boldsymbol{\mu}$ and $\mathcal{R}$ that determine the weights and the noncentralities of chi-squares. Therefore,

$$\Pr(\mathbf{Y}'\mathbf{Y} > t) = \Pr\left(\sum_{i=1}^{L} Y_i^2 > t\right) = \Pr\left(\sum_{i=1}^{L}\lambda_i\chi_{1,\gamma_i}^2 > t\right), \tag{6}$$

$$\gamma = \left\{\boldsymbol{\mu}'\,\mathbf{E}\left(\frac{1}{\sqrt{\boldsymbol{\lambda}}}\,\mathbf{I}\right)\right\} \circ \left\{\boldsymbol{\mu}'\,\mathbf{E}\left(\frac{1}{\sqrt{\boldsymbol{\lambda}}}\,\mathbf{I}\right)\right\}, \tag{7}$$

where the weights, $\boldsymbol{\lambda}$, are the eigenvalues of $\mathcal{R}$ and $\boldsymbol{\gamma}$ is the vector of non-centrality parameters.

The columns of the matrix $\mathbf{E}$ are orthogonalized and normalized eigenvectors of $\mathcal{R}$. The P-value for the statistic TQ = $\mathbf{Y}'\mathbf{Y}$ is obtained by setting $\boldsymbol{\mu}$ to zero and then calculating this tail probability at the observed value TQ = $t$. Note that the elements in $\mathcal{R}$, and therefore the eigenvectors, the eigenvalues $\lambda_i$, and the noncentralities explicitly depend on the $\beta$-coefficients through Eqs (2) and (5).

Our decorrelation approach uses a symmetric orthogonal transformation of the vector of statistics $\mathbf{Y}$ to a new vector $\mathbf{X}$, with the new joint statistic based on the sum of elements of $\mathbf{X}$, DOT = $\sum_{i=1}^{L} X_i^2$. The orthogonal transformation is defined as follows. Let $\mathbf{D} = \left( \frac{1}{\sqrt{\lambda}} \, \mathbf{I} \right)$ and define $\mathbf{X} = \mathbf{H}\,\mathbf{Y}$, where $\mathbf{H} = \mathbf{E}\,\mathbf{D}\,\mathbf{E}'$. The squared values, $X_i^2$, are one degree of freedom independent chi-square variables, thus DOT = $\mathbf{X}'\mathbf{X}$ is a chi-square random variable with $L$ degrees of freedom and noncentrality value of:

$$\gamma_c \;=\; \sum_{i=1}^{L} \gamma_i = \boldsymbol{\mu}' \mathcal{R}^{-1} \boldsymbol{\mu} = (\mathbf{H}\,\boldsymbol{\mu})'(\mathbf{H}\,\boldsymbol{\mu}). \tag{8}$$

The cumulative distribution of the new test statistic is thus,

$$\Pr(\mathbf{X}'\mathbf{X} > t) = \Pr(\chi^2_{L,\gamma_c} > t). \tag{9}$$

There are many ways to choose an orthogonal transformation, but a valid one for our purposes needs to have the following "invariance to order" property. Suppose we sample an equicorrelated MVN vector $\mathbf{Y}$ with a common correlation $\rho$ for all pairs of variables. Before decorrelating the vector, we permute its values to a different order. A permutation in this example is a legitimate operation, because an equicorrelation structure does not suggest a particular order of $\mathbf{Y}$ values. After an orthogonal transformation of $\mathbf{Y}$ to $\mathbf{X}$, the order of $\mathbf{X}$ entries may change due to permutation but their values should remain the same. Moreover, for the method to be useful in practice, we need the invariance to hold for a more general class of statistics than a simple sum of chi-squares, $\sum_{i=1}^{L} X_i^2$. For example, the Rank Truncated Product (RTP) is a powerful P-value combination method [12] that emphasizes small P-values: the RTP statistic $T_{\mathrm{RTP}}$ is the product of the $k$ smallest P-values, $k < L$, or equivalently, $T_{\mathrm{RTP}} = \sum_{i=1}^{k} [-\ln(P_i)]$, where $P_1 \leq P_2 \cdots \leq P_k$. Note that $-\ln(P_i)$ is no longer a one degree of freedom chi-square variable. Since DOT produces a set of independent one degree of freedom chi-squares, to use it with with RTP, one can convert the set of chi-squares to P-values and take the product of the first smallest values, which is the RTP statistic.

The "invariance to order" requirement implies that the value of DOT-statistic should not change due to a permutation of (equicorrelated) values in $\mathbf{Y}$. Not all orthogonal transformations meet the invariance to order criteria. It can be easily verified that neither the inverse Cholesky factor ($\mathbf{C}^{-1}$) transformation, $\mathbf{X} = \mathbf{C}^{-1}\,\mathbf{Y}$, nor another commonly used transformation $\mathbf{X} = \mathbf{E}\left( \frac{1}{\sqrt{\lambda}}\,\mathbf{I} \right)\mathbf{Y}$, have the invariance to order property, except in the special case of the sum of $L$ chi-squared variables $\sum_{i=1}^{L} X_i^2$. To clarify, we call this statistic "the special case," because, for example, in the case of RTP with $k = L$, the statistic $\sum_{i=1}^{L} -\ln(P_i)$ is no longer the sum of one degree of freedom chi-squares. Moreover, some transformations of equicorrelated data to independence, such as the Helmert transformation, may change values of $\mathbf{X}$ depending on the order of values in $\mathbf{Y}$, even in a special equicorrelation case of $\rho = 0$ (i.e., when variables in $\mathbf{Y}$ are independent). The proposed $\mathbf{H}$, as defined above, has both the invariance to order property and can be used with P-value transformations other than that to the one degree of freedom chi-square.

## Theoretical analysis of power

For exploration of power properties, it is useful to first consider the equicorrelation case, because in this case it is possible to derive illustrative equations that relate power to: (1) the number of SNPs, $L$; (2) the common correlation value for every pair of SNPs, $\rho$; and (3) the mean values of association statistics, $\boldsymbol{\mu}$. In the equicorrelation case, the correlation matrix can be expressed as $\mathcal{R}_\rho = (1 - \rho)\mathbf{I} + \rho\mathbf{11}'$. The eigenvalue vector of $\mathcal{R}_\rho$ has length $L$ but only two distinct values, $\boldsymbol{\lambda} = \{1 + \rho(L - 1), 1 - \rho, \ldots, 1 - \rho\}$.

For decorrelated statistic DOT, we derived a simple form of $L$ noncentralities by utilizing the Helmert orthogonal eigenvectors [81, 82] as follows:

$$\delta_1 \quad = \quad \frac{\left(\sum_{i=1}^{L} \mu_i\right)^2}{L(1 + (L - 1)\rho)} = \frac{L\bar{\boldsymbol{\mu}}^2}{1 + (L - 1)\rho}, \tag{10}$$

$$\delta_{j>1} \quad = \quad \sum_{i=1}^{j-1} \frac{(\mu_i - \mu_j)^2}{L(1 - \rho)}, \tag{11}$$

where $\bar{\boldsymbol{\mu}}$ is the average of the values in $\boldsymbol{\mu}$. Next, let

$$\delta_s = \sum_{j=2}^{L} \delta_j = (L - 1)\frac{\bar{d}}{2(1 - \rho)}, \tag{12}$$

where $\bar{d}$ is the average of $d_{ij} = (\mu_i - \mu_j)^2$, over all pairs of $\mu_i$ and $\mu_j$, such that $i < j$. The values in $d_{ij}$ are the pairwise squared differences in the standardized effect values as captured by the vector $\boldsymbol{\mu}$. This representation yields the noncentrality of DOT as a function of the common correlation and the mean standardized effect size as:

$$\gamma_c = \frac{L\bar{\boldsymbol{\mu}}^2}{1 + (L - 1)\rho} + \delta_s. \tag{13}$$

Note that as $L$ increases, the first term in Eq (13) approaches $\bar{\boldsymbol{\mu}}^2/\rho$, while the sum of the remaining noncentralities, $\delta_s$, increases linearly with $L$, as long as the average of the squared effect size differences, $\bar{d}$, does not depend on $L$. Thus, the noncentrality of the decorrelated statistic DOT is expected to steadily increase with $L$ and become approximately $\bar{\boldsymbol{\mu}}^2/\rho + (L - 1)\frac{\bar{d}}{2(1-\rho)}$.

Next, we consider the distribution of the statistic TQ = $\mathbf{Y}'\mathbf{Y}$. Note that $\sum_{i=1}^{L} \delta_i = \sum_{i=1}^{L} \gamma_i$, where $\gamma_i$'s are the noncentralities for TQ and $\delta_i$'s are the noncentralities of DOT. In the equicorrelation case, the distribution TQ reduces to the weighted sum of two chi-square variables, because there are only two distinct eigenvalues that correspond to $\mathcal{R}_\rho$, namely:

$$\Pr(\mathbf{Y}'\mathbf{Y} > t) \quad = \quad \Pr\{(1 + (L - 1)\rho)\chi^2_{1,\gamma_1} + (1 - \rho)\chi^2_{L-1,\gamma_c-\gamma_1} > t\} \tag{14}$$

$$= \quad \Pr\left\{\chi^2_{1,\gamma_1} + \frac{1 - \rho}{1 + (L - 1)\rho}\chi^2_{L-1,\gamma_c-\gamma_1} > \frac{t}{1 + (L - 1)\rho}\right\}. \tag{15}$$

The term $\frac{1-\rho}{1+(L-1)\rho}\chi^2_{L-1,\gamma_c-\gamma_1}$ in Eq (15) approaches the constant $\frac{\bar{d}(1-\rho)}{2\rho^2} + \frac{1-\rho}{\rho}$ as $L$ increases. Therefore, under the null hypothesis, the distribution of the quadratic form $\mathbf{Y}'\mathbf{Y}$ can be well approximated by the location-scale transformation of the one degree of freedom chi-squared random

variable:

$$\Pr\left\{\frac{\mathbf{Y}'\mathbf{Y} - (L-1)(1-\rho)}{(L-1)\rho + 1} > \chi^2_\alpha\right\} \approx \alpha, \tag{16}$$

where $\chi^2_\alpha$ is $1 - \alpha$ quantile of the one degree of freedom chi-square distribution.

To summarize, we just showed that the distribution of the decorrelated set of variables gains in the total noncentrality with $L$, while the distribution of the sum $\mathbf{Y}'\mathbf{Y}$ depends heavily only on the noncentrality of the first term, $\gamma_1$. The approximate power of the test based on the statistic TQ = $\mathbf{Y}'\mathbf{Y}$ can be computed as:

$$\Pr(\text{TQ} > t) \quad \approx \quad 1 - \Psi(t), \tag{17}$$

$$t \quad = \quad \chi^2_\alpha + \frac{1-\rho^*}{\rho^*} + \frac{1}{2}\frac{(1-\rho^*)\bar{d}}{(\rho^*)^2}, \tag{18}$$

where $\rho^* = \sqrt{\overline{\rho^2_{ij}}}, \mu^* = (|\bar{\boldsymbol{\mu}}|)^2$ and $\Psi(\cdot)$ is a one degree of freedom chi-square CDF with the noncentrality $L\mu^*/((L-1)\rho^* + 1)$, evaluated at $t$. The ceiling noncentrality value $\gamma^*$, as $L \to \infty$, is thus

$$\gamma^* \quad \approx \quad \mu^*/\rho^*. \tag{19}$$

Let us re-emphasize the point that a test based on the distribution of the TQ statistic is expected to be less powerful than DOT in the presence of heterogeneity among effect sizes. Heterogeneity in LD will contribute to the difference in power. Starting with an equicorrelation model, we can introduce perturbations to the common value, $\rho > 0$, by adding noise derived from a rank-one matrix $\mathbf{U}\,\mathbf{U}'$, where $\mathbf{U}$ is a vector of random numbers. Specifically, perturbations can be added as $\mathbf{B} = \mathcal{R}_\rho + \mathbf{U}\mathbf{U}'$. Next, $\mathbf{B}$ should be standardized to correlation as $\mathbf{B}_R = \{\sqrt{1/\text{Diag}(\mathbf{B})}\,\mathbf{I}\}\,\mathbf{B}\{\sqrt{1/\text{Diag}(\mathbf{B})}\,\mathbf{I}\}$. When elements in $\mathbf{U}$ are close to zero, the matrix $\mathbf{B}_R$ deviates from $\mathcal{R}_\rho$ by only a small jiggle around $\rho$. Matrix $\mathbf{B}_R$ provides a way to construct random correlation matrices in a controlled manner, where the degree of departure from the equicorrelation is controlled via the range of the elements in $\mathbf{U}$. The utility of $\mathbf{B}_R$ is that it represents a perturbation of $\mathcal{R}_\rho$, and we expect our power results under equicorrelation case to hold approximately, at least for small jiggles around $\rho$. Nevertheless, it turns out that even for a more general correlation structure, our power approximations still hold, which we show via extensive simulation studies.

## LD patterns from the 1000 Genome Project

In a separate set of simulation experiments, we utilized realistic LD patterns using data from the 1000 Genomes Project [83]. For every simulation experiment, we selected a random set of consecutive SNPs from a chromosome 17 region, that was spanning over 100 Kb and included SNPs from the gene *FGF11* to the gene *NDEL1*. There was no particular reason for choosing this chromosome, but we expect our results to be generalizable to other regions of the genome in the sense that LD structure among SNPs on chromosome 17 is representative of LDs throughout the genome. Perhaps more important, and a potential limitation of our simulations, is the choice of the association model. That is, the model assumed high heterogeneity in effect sizes and statistics were combined for only proxy SNPs (those SNPs with zero effect sizes). Each stretch of consecutive SNPs contained from 10 to 200 SNPs with the minimum allele frequency 0.025. A random portion of SNPs in every set carried no effect on the outcome

on its own, and we considered these SNPs to be proxies for causal variants due to LD. The median LD correlation varied from approximately -0.6 to 0.98 between random stretches of SNPs. The number of proxy SNPs varied from 3 to 197 across simulations. The sample size was also set to be random and varied from 500 to 3000 across simulations. Effect sizes for causal variants were modeled by $\beta$-coefficients, as given by Eq (1), and drawn randomly from the interval [-0.4, 0.4]. Different combinations of the number of causal SNPs, their individual effect sizes and LD patterns among them resulted in total proportion of phenotypic variance explained (i.e., the multiple correlation coefficient) varying from $10^{-5}$% (fifth percentile) to 7% (ninety-fifth percentile) with the mean value of 2.5% and the median value of 1%. Summary statistics were sampled from the MVN distribution with parameters given by Eqs (2) and (4). To check the validity of our approach of sampling the summary statistics directly, we first conducted a separate set of extensive simulation experiments, in which power and type-I error rates were obtained by simulating individual data and then TQ and DOT statistics were computed by running the actual regression analysis. We confirmed excellent agreement between the two approaches, thus most of the subsequent simulations were conducted by sampling the summary statistics directly (these results are not shown here).

## Author Contributions

**Conceptualization:** Olga A. Vsevolozhskaya, Dmitri V. Zaykin.

**Data curation:** Min Shi, Dmitri V. Zaykin.

**Formal analysis:** Olga A. Vsevolozhskaya, Dmitri V. Zaykin.

**Methodology:** Olga A. Vsevolozhskaya, Dmitri V. Zaykin.

**Software:** Dmitri V. Zaykin.

**Writing – original draft:** Olga A. Vsevolozhskaya, Dmitri V. Zaykin.

**Writing – review & editing:** Min Shi, Fengjiao Hu.

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
