## [Decision Letter · Decision Letter 0]

25 Oct 2019

Dear Dr Zaykin,

Thank you very much for submitting your manuscript 'DOT: Gene-set analysis by combining decorrelated association statistics' for review by PLOS Computational Biology. Your manuscript has been fully evaluated by the PLOS Computational Biology editorial team and in this case also by independent peer reviewers. The reviewers appreciated the attention to an important problem, but raised some substantial concerns about the manuscript as it currently stands. While your manuscript cannot be accepted in its present form, we are willing to consider a revised version in which the issues raised by the reviewers have been adequately addressed. We cannot, of course, promise publication at that time.

Sincerely,

Jennifer Listgarten

Associate Editor

PLOS Computational Biology

Thomas Lengauer

Methods Editor

PLOS Computational Biology

[LINK]

Reviewer's Responses to Questions

**Comments to the Authors:**

Reviewer #1: In this paper, the authors present a new summary-statistics-based method for testing a group of common SNPs in aggregate for association to a phenotype. Unlike previous approaches, the authors' test statistic explicitly (and exactly) removes correlation between the individual SNPs' summary statistics.

I generally like this paper and appreciate the authors' precision and rigor in deriving and presenting their method. Their theoretical results concerning the power of their test as well as others are also a valuable contribution. So I generally feel this is a very solid contribution to the field. In the long-term I would suggest that the authors consider applications of their framework beyond set-testing since my impression is that the growing number of highly significant associations between *individual* SNPs and phenotypes will eventually cause set-testing to decline as an approach in the common-variant realm. But this is beyond the scope of this paper and for now there remains a substantial community of users of set tests who could benefit from the approach described by the authors.

Regarding the technical substance of the paper, I have the following major comments:

- I'm unclear on the phenomenon whereby TQ tests don't experience an increase in power as more SNPs are added to the model, e.g., in Setting 1. Looking at the authors' model, in which the variance of the environmental noise, epsilon, is set at 1, it would seem that the more SNPs I add to the model with non-trivial effects, the more phenotypic variance is produced by the genetics. In the limit of infinite SNPs and constant-magnitude environmental noise then, the phenotype should be deterministically set by genotype. It would seem unintuitive that in this situation the TQ tests wouldn't have full power. What am I missing? Are the authors scaling something somewhere?

- Relatedly, it would help if the authors included in their methods section more detailed descriptions of their simulation set ups especially including sample size and proportion of phenotypic variance explained by genotype for each simulation (including the simulations with real genotypes).

- I don't know if the proportion of variance explained by genotype is high in the authors' simulations. But if it is, do they expect their results to generalize to settings where this is not the case? For real traits, any one set of tens to hundreds of contiguous SNPs typically only explains a very small proportion (on the order of 1%, usually even less than that) of phenotypic variance, so I'd be interested to see if this is the case in the simulations here. Sometimes it's okay to simulate small sections of genome explaining high proportions of phenotypic variance as long as sample size is lowered in some corresponding way, but if this is the case here the authors should explain and perhaps use their theory to justify.

- How do the authors expect their statistic to behave in the presence of near-perfect LD? It seems they don't regularize their LD matrix, which surprised me. I would be interested to see power results under a simulation setting where two SNPs, only one of which is causal and contains 75% of the causal signal in locus, have a) 99% correlation and b) 100% correlation.

- For the simulations with real genotypes, how was the 100kb region on chromosome 17 chosen? Do the authors expect the simulation results to generalize to other regions of the genome as well? If they are unsure, is it computationally feasible to do simulations where random sets of contiguous SNPs are chosen from the whole genome?

- How were the genes ESR1, FGFR2, RAD51B, and TOX3 chosen by the authors for demonstration of their method? Does this set include all the genes found in the Min et al paper to have association with breast cancer? Would it be possible to test a larger set of genes chosen more systematically so that readers can have a sense for whether the authors' approach should in general be preferred over other approaches? Or perhaps to test a few genes chosen by authors of other set testing methods papers?

- Do the authors think it would make sense to compare (either in simulation or in practice) to the gene-level test in de Leeuw 2016 PLOS Comp Bio since that method also provides a way to test the SNPs surrounding an individual gene for association while accounting for correlation between variants in order to boost power? Relatedly: ACAT seems to be a method intended primarily for testing of rare variants in sequence data; could it be that this makes it an inappropriate comparison point?

- I liked the way the authors argued for their particular choice of pseudoinverse by suggesting that exchangeability of SNPs should be preserved by this operation. Kudos!

I also have the following minor comments:

- It seems that the claims about the scaling of power as a function of L are for fixed rho > 0, because when rho=0 the tests considered are equivalent. The authors may want to clarify this.

- In the definition of r_ij on page 2, should there be a square-root in the denominator?

- On page 3 there is a typo in "This general idea is straightforward and HAVE been used..." (emphasis mine)

- What was the sample size of the breast cancer data set that the authors analyzed?

- In Equations 4 and 5, rho_ij appears on both sides of the equations.

- The derivation of the covariance matrix of the vector of summary statistics can be carried out without the delta method but under the assumption of Gaussian genotypes (which is justifiable for large sample size and MAF bounded away from zero). See Proposition 2 in the supplement of Reshef et al 2018 Nat Genet. The authors may wish to comment on whether these two derivations give different results and if so why not.

- For the results in Table 6: 1) which set of genotypes were the phenotypes simulated from? 2) Which set of genotypes was used as the reference panel? The only genotypes I saw mentioned were 1000 Genomes, but two distinct sets of genotypes are required for the described analysis.

Reviewer #2: Zaykin et al propose DOT, a new method for Gene Based Association Testing. There is demand for a gene (or set-based) method, so a method that improves upon previous methods would be of much interest and (with easy to use software) could become highly used. Zaykin perform many simulations to show that DOT has the potential to improve on a state-of-the-art method, VEGAS (and also ACAT, a method I am not familiar with). They also have a real data example, but this is very limited. While I am not convinced from this draft alone, I believe that by including an extra simulation method, and a more convincing application, DOT could be a useful addition to the field.

Major points

Reading the method (and apologies that I did not understand all the details), DOT appears similar to methods which first compute principal components for each gene (ie eigen decompose the snp snp correlation matrix), then regress the phenotype on these (consider the following paper, or derivatives https://onlinelibrary.wiley.com/doi/pdf/10.1002/gepi.20219). Thus I require convincing this method is different to / an improvement on those.

The format of the paper makes it challenging to read. Usually methods would come before results. However, if the journal requires such a style, then you must give some brief details at the start of results.

I consider there to be insufficient detail of the simulations. For example, I can't see sample size and rho was hard to find. Is it the case for all simulations that all L snps are assigned effects, or just the first one?

It is good you compare with vegas (TQ?). But to my knowledge, the most common methods are SCAT, or magma, and my preferred is Fast-LMM-Set, so would ideally like at least one of these considered (or a statement with justification that these very similar to VEGAS)

The application is very limited. While I appreciate there is justification for the choice, unfortunately it looks odd to consider only four handpicked loci, rather than perform a genome wide analysis.

I believe you require odds ratios for the SNPs in table 8 (ideally from multi snp analysis and perhaps those from single snp)

Minor Points

I applaud the range of simulations, and also of considering situations where DOTS is not well-suited

I also like the insight into how DOT has the potential to gain power (when a wide spectrum of effect sizes, which is thought likely to be the case with complex traits).

In the simulations, it is hard to understand the effect sizes. Can you instead report in terms of heritability, ideally both (average) phenotypic variance explained by the gene/region, and (average) variance explained by most significant individual snp

The tables (and I think figures) require captions. In generally, these should give a full description (or if the same, say "see Table 1... etc"), rather than relying on the user to parse through the main text.

Good that a github page is provided with software (although I have not tested)

Please provide a summary of run time for a decent sized analysis.

Very Minor Points

Intro; It is important to distinguish situations ... I suggest you replace second "in which" with "from those" or something similar

I would prefer if you provided more thresholds when testing the false positive rates (e.g. show not just alpha 1e-4, but also say 0.05, maybe a few others, in supplement if necessary)

It is good you can accommodate covariates, but is this feature used in application?

Signed Doug Speed

Reviewer #3: In this manuscript, the authors combined single-SNP summary statistics in order to conduct joint analysis of a set of SNPs without accessing original genotype-phenotype datasets. To develop efficient overall summary-statistic, the authors used a decorrelation trick To simplify the correlation structure of the the vector of the single-SNP summary-statistics. The later are correlated by construction. Thus, by rotating the this vector over the eigenvectors of its corresponding correlation matrix one can simplify its correlation structure. Although the decorrelation-trick of a response vector is not a new concept—it has been used for kinship matrix several times in linear mixed models in presence of familial data, e.g. FastLMM— the theoretical and analytical development of the DOT p-values in this manuscript is relevant, in the context summary-statistic association.

Major and Minor Comments are dteailed in a PDF file attached to this review.

**Have all data underlying the figures and results presented in the manuscript been provided?**

Reviewer #1: Yes

Reviewer #2: Yes

Reviewer #3: Yes

PLOS authors have the option to publish the peer review history of their article (what does this mean?). If published, this will include your full peer review and any attached files.

Reviewer #1: No

Reviewer #2: Yes: Doug Speed

Reviewer #3: No

---

## [Decision Letter · Decision Letter 1]

25 Feb 2020

Dear Dr Zaykin,

Thank you very much for submitting your manuscript "DOT: Gene-set analysis by combining decorrelated association statistics" for consideration at PLOS Computational Biology. As with all papers reviewed by the journal, your manuscript was reviewed by members of the editorial board and by several independent reviewers. The reviewers appreciated the attention to an important topic. Based on the reviews, we are likely to accept this manuscript for publication, providing that you modify the manuscript according to the review recommendations, and in particular those of reviewer #1.

Sincerely,

Jennifer Listgarten

Associate Editor

PLOS Computational Biology

Thomas Lengauer

Methods Editor

PLOS Computational Biology

[LINK]

Reviewer's Responses to Questions

**Comments to the Authors:**

Reviewer #1: Overall the authors have addressed my theoretical and methods-related concerns quite well in this revision.

However, I still have serious reservations about the authors' analysis of real data, which analyses a very small set of genes that were not chosen systematically. I previously wrote: "Would it be possible to test a larger set of genes chosen more systematically so that readers can have a sense for whether the authors’ approach should in general be preferred over other approaches? Or perhaps to test a few genes chosen by authors of other set testing methods papers?"

The authors did not perform this analysis, and so I still do not know whether their method is more powerful than existing methods beyond the very small set of genes they have analyzed. (The addition in revision of a second phenotype, cleft lip, analyzed in the same way as the first phenotype did not give me a better global sense for why people should use this method.) My understanfing of what the authors have shown is that:

a) DOT assigns lower p-values than other methods do to the 4 selected breast cancer genes. This seems weak to me first because lower p-values don't necessarily correspond to higher power (a method can give very low p-values on 1 % of alternatives but fail to reject the null the rest of the time), and second because these genes have already been prioritized by other methods, suggesting that their connection to breast cancer is not a new discovery enabled by DOT. For example, these genes seem from the text to harbor previously reported risk SNPs. Am I missing something?

b) DOT can point at new SNPs associated with breast cancer and cleft lip at these known loci (Tables 10 and 12). But the authors also state (appropriately) that since these results don't come with p-values they should be interpreted with caution, and they also state that cannot conclude that these SNPs are causal but rather only additional proxy SNPs. So I'm unsure what we can confidently learn from these results.

I personally don't find (a) or (b) to be strong reasons that practitioners should use DOT.

Overall, I see two ways forward:

1. The authors can carry out a systematic analysis of the performance of their method on real data. For example, they could run the method on a larger set of genes (e.g., all protein coding genes, or all genes expressed in breast tissue, or a set of genes benchmarked in other set testing papers). This would allow the authors to say things like "in a systematic analysis, our method identified X genes to be in loci that are significantly associated with breast cancer, while competing methods identified only Y such genes." I think this would make a much stronger case for the use of this method. And if it's not true, then that is important for potential users to know even if it doesn't preclude publication of the paper.

2. Alternatively, recognizing they have performed extensive revisions already, the authors can add a statement explaining that the genome-wide performance of their method is yet-uncharacterized and would be important to assess in future work.

I suppose it would be okay to publish the paper in case 2, but my opinion is that I would be less excited about it. Not answering the central question of whether DOT is more powerful than other methods in practice on real data is not consistent with the otherwise high level of statistical rigor in this potentially interesting paper.

Minor comments:

- Just above Table 1, you have a typo: "the column labeled \\hat\\gamma provide the average noncentrality value" ("provide" should be "provides")

- In the sentence “Different combinations of sample size, the number of causal SNPs, their individual effect sizes and LD patterns among them, resulted in total proportion of phenotypic variance explained...", whose addition I appreciate in this revision, sample size should not be enumerated as one of the parameters that affects the total proportion of phenotypic variance explained.

- On page 10, you cite "Min et al. [27, 28]" but neither of refs. 27 or 28 has Min as the last name of a first author in your bibliography.

- In your response to R1.1.6, you state that eqns 22 and 27 in Reshef et al. 2018 are derived under the null, but this is not true: Eq 22 defines the computation of summary statistics from data (regardless of model) and Equation 27 includes a parameter beta which can be non-zero. A question therefore remains about the relationship between your derivation and the derivation that assumes Gaussian genotypes. (Fine if you want to drop this issue.)

Reviewer #2: The authors have made a careful response and I am happy with the changes.

Reviewer #3: No addtional comments

**Have all data underlying the figures and results presented in the manuscript been provided?**

Reviewer #1: Yes

Reviewer #2: Yes

Reviewer #3: None

PLOS authors have the option to publish the peer review history of their article (what does this mean?). If published, this will include your full peer review and any attached files.

Reviewer #1: No

Reviewer #2: Yes: Doug Speed

Reviewer #3: No
---

## [Decision Letter · Decision Letter 2]

23 Mar 2020

Dear Dr Zaykin,

We are pleased to inform you that your manuscript 'DOT: Gene-set analysis by combining decorrelated association statistics' has been provisionally accepted for publication in PLOS Computational Biology.

Best regards,

Jennifer Listgarten

Associate Editor

PLOS Computational Biology

Thomas Lengauer

Methods Editor

PLOS Computational Biology

Reviewer's Responses to Questions

**Comments to the Authors:**

Reviewer #1: I thank the authors for their revision, and I am happy to recommend acceptance given the clarifications the authors made about their analysis of real data.

Setting aside this one point of disagreement, I feel this is very high quality work and I commend the authors on their valuable contribution to the field.

**Have all data underlying the figures and results presented in the manuscript been provided?**

Reviewer #1: Yes

PLOS authors have the option to publish the peer review history of their article (what does this mean?). If published, this will include your full peer review and any attached files.

Reviewer #1: No

---

## [Editor Report · Acceptance letter]

6 Apr 2020

PCOMPBIOL-D-19-01433R2 

DOT: Gene-set analysis by combining decorrelated association statistics

Dear Dr Zaykin,

I am pleased to inform you that your manuscript has been formally accepted for publication in PLOS Computational Biology. Your manuscript is now with our production department and you will be notified of the publication date in due course.

With kind regards,

Matt Lyles
